# Look, Then Speak: Social Tokens for Grounding LLMs in Visual Interactions

**Kathy Garcia**[1]*, **Vighnesh Subramaniam**[2]*, **Boris Katz**[2],
**Brian Cheung**[2]
[1]Department of Cognitive Science, Johns Hopkins University [2]MIT CSAIL, CBMM
[1]kgarci18@jhu.edu
[2]{vsub851, boris, cheungb}@mit.edu

## Abstract

Social interactions remain a major challenge for large language models (LLMs), which struggle to incorporate visual context and social cues. We propose social tokens, a lightweight mechanism that introduces socially grounded visual information into a frozen LLM. To construct these tokens, we first fine-tune a visual encoder on videos of social interactions to learn embeddings that capture socially relevant cues. A small MLP then projects these embeddings into the LLM's embedding space, where they are inserted into the input sequence as local and global summaries of the scene. This representational alignment enables the LLM to condition generation on social context without updating its parameters. Empirically, social tokens substantially reduce perplexity on social dialogue and caption datasets, improve alignment with human social judgments, and receive high attention weights during socially salient segments, underscoring both their utility and interpretability.

## 1 Introduction

Human language is an inherently social tool [1]. Humans interpret language effortlessly using visual, auditory, and situational context with the goal of social communication [16, 20]. By contrast, large LLMs are trained solely on text corpora, without visual, auditory, or situational context, yet are expected to integrate socially with humans in practical tasks.

Currently, most efforts at social alignment happen post-training via text-only methods. Techniques such as reinforcement learning from human feedback (RLHF) [4, 18] or video feedback (RLVR) [8, 12] can tune models on abstract social objectives, but social interactions often require grounding in rich multimodal stimuli like tone of voice, facial expressions, and body language. Current LLMs are ill-equipped for these stimuli [9, 6, 21]. Furthermore, even multimodal LLMs like vision-language models (VLMs) are trained without these cues and suffer from similar problems. Recent studies show that VLMs often ignore visual context, and some even argue that visual grounding provides little benefit at scale [5, 24].

In this work, we investigate social representational alignment in LLMs using social tokens. We augment LLMs to have access to social interactions using finetuned encoders from other modalities like vision or audio. We inject social tokens using classification embeddings from the encoder, projected via a multilayer perceptron (MLP). After injecting social tokens, we find that (1) next-word prediction improves in settings where social interactions and multimodal cues are important such as with social multi-turn dialogue [2] or video captions of social scenes [9], and (2) social tokens are interpretable and useful, receiving large amounts of attention in the underlying language model.

---

*Equal Contribution, Corresponding authors.

39th Conference on Neural Information Processing Systems (NeurIPS 2025) Workshop: 3rd Edition of the Workshop on Unifying Representations in Neural Models.

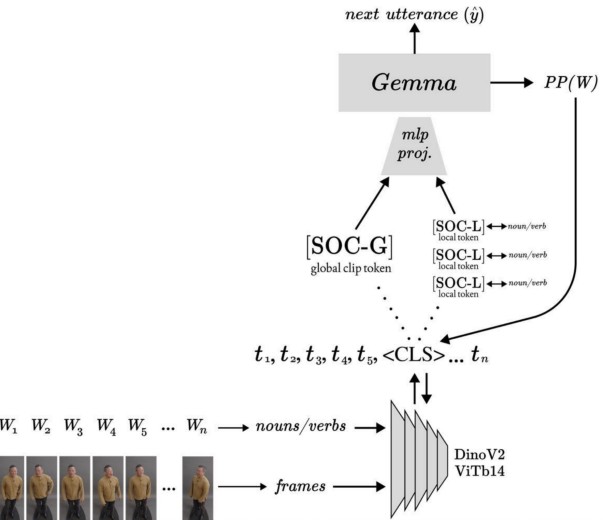

Figure 1: **Injecting social understanding into LLMs via social tokens**. Social tokens are projected embeddings produced by modality-specific encoders (e.g., vision; extensible to audio) and inserted into an LLM (e.g., Gemma). Given a video and its time-aligned transcript, we POS-tag the text to select nouns and verbs using the spaCy parser [10], retrieve the temporally nearest frame for each selected word, and encode it with DINOv2. The frame `[CLS]` embedding is projected by a learned MLP into the LLM's embedding space to form a local social token `[SOC-L]`, which is inserted immediately after the corresponding text token. A global token `[SOC-G]` is computed by averaging the local token vectors and is prepended to the sequence.

## 2 Related Work

**Social Interactions in LLMs and VLMs**   Humans seamlessly interpret language using visual, vocal, and situational cues. In contrast, LLMs often lack robust social understanding [11, 9, 3], in part because they are trained on text alone in disembodied settings, leading to misinterpretations and bland responses to socially nuanced inputs. Benchmarks targeting social understanding confirm that even large LLMs struggle, and these shortcomings extend to VLMs [7], suggesting that image–text pretraining alone is insufficient. Moreover, scaling data and model size does not reliably improve social understanding; representational similarity analyses even find BERT-based models strongest on certain social signals [9]. Our work tackles these gaps by introducing *social tokens*—modality-derived vectors explicitly augmented for social understanding—and injecting them into LLMs/VLMs.

**Vision Language Model Training**   Prior work augments language models with visual information to build VLMs, typically using pretrained, frozen vision encoders and a learned projector to inject visual tokens into their embedding space, improving tasks such as image captioning and visual question answering [14, 23, 13]. However, more recent studies show that VLMs often ignore them, and at scale, visual grounding has little effect on language model performance [5, 25, 24]. Our setting is different: social interaction is a rich, sometimes unpaired multimodal domain where LLMs and VLMs show deep gaps. Hence, we argue that effective augmentation requires cross-modal translation and encoders that are socially aligned.

## 3 Methods

We introduce *social tokens*: learned vectors derived from video frames, that are inserted into a frozen LLM to improve reasoning about social interactions and relations. Our approach adapts standard VLM training methods while modifying the interface between a visual encoder and LLM to better integrate social cues.

**Setup and notation**   Let $L$ be our large language model and $V$ be a visual encoder (e.g. DINOv2 [17] or CLIP [19]). We assume we can extract inputs from our video frames, which we refer to

as $\mathcal{F} = F_1, \ldots, F_m$ and a time-aligned transcript $\mathcal{W} = W_1, \ldots, W_n$ that together depict social interactions. Tokenizing the transcript with $L$'s tokenizer yields $t_1, \ldots, t_k$.

Our goal is to use $V$ to build visual embeddings that are *socially aligned*. We take these embeddings, project via an MLP, and concatenate them with the token representations in $L$ as *social tokens*. We tune the encoder $V$ and projector to align the representations across the two models and improve multimodal understanding for social interactions ( Figure 1). Our design is flexible and can be extended to further modalities like audio.

**Finetuning the visual encoder** We begin by finetuning our visual encoder $V$ on the frames $\{F_1, \cdots, F_m\}$ from our dataset using a self-supervised reconstruction task such as the DINO loss. This step encourages $V$ to encode fine-grained social cues that will be useful downstream. These finetuned `[CLS]` embeddings are what we refer to as social tokens: representations that are more closely aligned to social interactions. For each frame $F_j$, we take the social token as a frame-level visual representation $v_j$. We then finetune DINOv2 using the teacher–student self-distillation objective [17].

**Local and global social tokens** From the transcript, we extract nouns and verbs via POS tagging, following prior work showing that these parts of speech contain rich information for social scene understanding [9]. Restricting to nouns and verbs encourages social alignment and mitigates common VLM failures such as overlooking social context [5]. For each such word occurrence at time $\tau$, we select the nearest frame $F_j$ and compute its embedding $v_j$. An MLP projector $g(\cdot)$ maps $v_j$ into $L$'s embedding space, yielding a *local* social token $\mathtt{[SOC\text{-}L]}_p = g(v_j)$ associated with token position $p$ of the noun/verb. We define the *global* social token as the average of local tokens in the utterance:

$$\mathtt{[SOC\text{-}G]} = \frac{1}{|\mathcal{P}|} \sum_{p \in \mathcal{P}} \mathtt{[SOC\text{-}L]}_p \tag{1}$$

We form the input sequence by prefixing $\mathtt{[SOC\text{-}G]}$ and inserting each $\mathtt{[SOC\text{-}L]}_p$ immediately after its corresponding text token:

$$t_p : \quad \big\{ \mathtt{[SOC\text{-}G]}, \, t_1, \, t_2, \, \ldots, \, t_p, \, \mathtt{[SOC\text{-}L]}_p, \, \ldots, \, t_k \big\} \tag{2}$$

**Training objective and inference** We freeze $L$ and optimize the parameters of $V$ and the projector $g$ only. Given the augmented sequence, we apply standard next-token cross-entropy over *text* tokens; losses for the inserted $\mathtt{[SOC\text{-}*]}$ tokens are masked out. Gradients flow through the inserted social tokens into $g$ and $V$, aligning visual embeddings with $L$'s token space.

At test time, we repeat noun/verb extraction, frame selection, projection, and token insertion, and then decode with $L$. The design is modality-agnostic and can be extended by adding an audio encoder and projector in parallel.

## 4   Experiments

We evaluate our approach for improving social understanding and representational alignment in LLMs using social tokens.

**Models and Datasets** For all experiments, we use the `DINOv2` visual encoder and `Gemma-2-2b-it` [22] as the LLM. We finetune DINOv2 and our social token projection on two datasets: (1) the Seamless Interaction Dataset [2], which contains multi-turn dialogues with facial expressions and 2-minute video clips with transcripts, and (2) odd-one-out tasks [9] drawn from 250 3-second social videos in Moments in Time [15]. We first finetune DINOv2 on Seamless frames to create social tokens, then use frames and transcripts to train the projection as described above.

**Evaluation** We evaluate by measuring perplexity on held-out Seamless transcripts and odd-one-out captions, comparing LLMs with and without social tokens. We further analyze interpretability with Gemma-2 attention maps to assess how the model leverages social tokens.

## 5   Results

**Transcript Perplexity** We first measure whether social tokens lead to a general improvement in an LLM's ability to predict social dialogue. We calculate the perplexity of generated responses from the

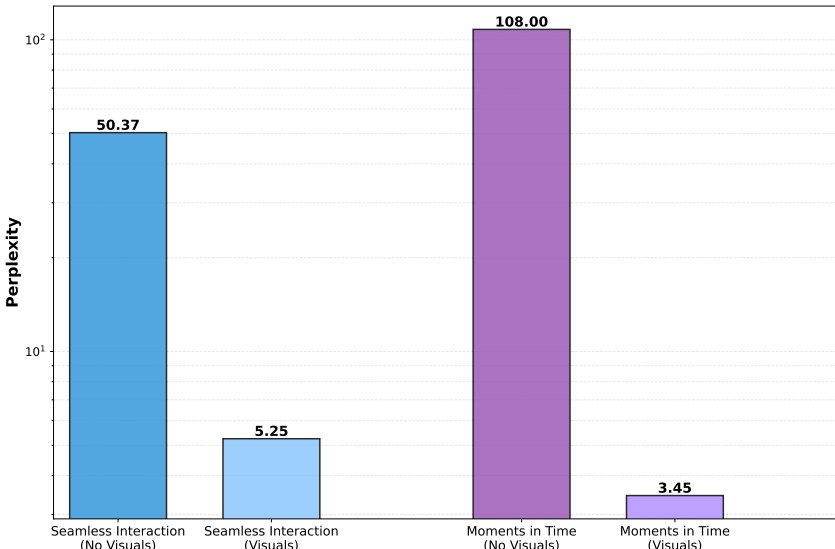

Figure 2: **Social tokens improve next-word prediction.** We measure perplexity of generated predictions from Gemma-2, tuned with social tokens and without social tokens. The perplexity is measured on a held-out dialogue set from the Seamless Interaction dataset (left) and a held-out caption set from the odd-one-out task (right). We find that including social tokens leads to a dramatic improvement in the model's ability to predict further social tokens.

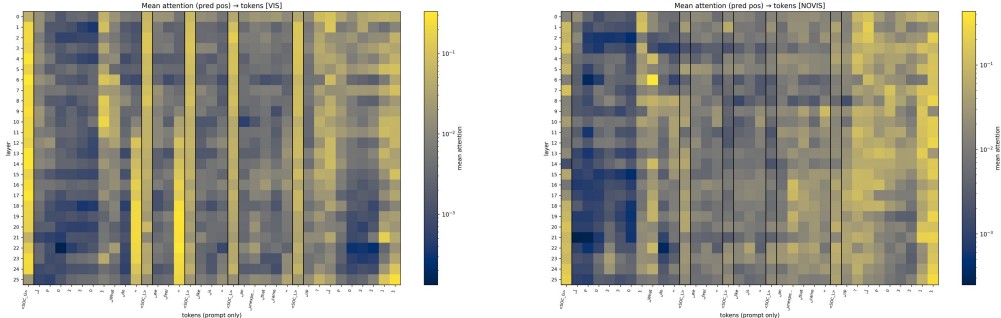

Figure 3: **Attention Scores of Social Tokens**: We analyze the attention maps from attention layers to understand how much attention is given to social tokens. On the left, we show the results from a string of tokens while on the right, we include a baseline where social tokens are replaced with zero vectors to understand whether there is a positional bias. We find that global social tokens receive a large amount of attention and this is not due to a positional bias as seen from the right visualization.

LLM for a set of held-out dialogues from the Seamless Interaction dataset. This set is not included in training. We extract social tokens following the previously described procedure, aligning based on nouns and verbs.

Results are shown in Figure 2. Incorporating social tokens yields a substantial reduction in perplexity compared to text-only inputs, indicating a clear improvement in model performance. Dialogue in the Seamless Interaction dataset is challenging to model, yet visual cues from social tokens yield substantial improvements.

**Attention Analysis of Social Tokens**    To assess the contribution of social tokens to model behavior, we analyze attention distributions (Figure 3). Attention heatmaps reveal that both local and, in particular, global social tokens receive substantial weight from the LLM during prediction. These results show that the model actively leverages social tokens, which play a meaningful role in guiding downstream performance.

# 6    Conclusion

In this work, we introduced social tokens, a new mechanism designed to improve LLM performance on social tasks by aligning socially informative visual encoders with language models. This approach yielded consistent improvements in social understanding across socially relevant datasets and enhanced alignment with human judgments. Future work will include deeper ablations and extension to other modalities (e.g., audio) to broaden performance gains.

## Acknowledgements

This work was supported by the Center for Brains, Minds, and Machines, NSF STC award CCF-1231216, the Brains, Minds, and Machines Summer School, the NSF award 2124052, the NSF GRFP DGE-2139757, the MIT CSAIL Machine Learning Applications Initiative, the MIT-IBM Watson AI Lab, the CBMM-Siemens Graduate Fellowship, the DARPA Mathematics for the DIscovery of ALgorithms and Architectures (DIAL) program, the DARPA Knowledge Management at Scale and Speed (KMASS) program, the DARPA Machine Common Sense (MCS) program, the Department of the Air Force Artificial Intelligence Accelerator under Cooperative Agreement Number FA8750-19-2-1000, and the Air Force Office of Scientific Research (AFOSR) under award number FA9550-21-1-0399. The views and conclusions contained in this document are those of the authors and should not be interpreted as representing the official policies, either expressed or implied, of the Department of the Air Force or the U.S. Government. The U.S. Government is authorized to reproduce and distribute reprints for Government purposes notwithstanding any copyright notation herein. V.S. is supported by the National Science Foundation Graduate Research Fellowship under Grant No. 2141064. Any opinion, findings, and conclusions or recommendations expressed in this material are those of the authors and do not necessarily reflect the reviews of the National Science Foundation.

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
