# OpenReview forum: "Look, Then Speak: Social Tokens for Grounding LLMs in Visual Interactions"
_NeurIPS.cc/2025/Workshop/UniReps — UniReps2025_

### Official Review · Reviewer_93Kz · 2025-09-12
**Encoder to project visual features as social tokens to be combined with text tokens, offering performance gains in social interaction settings.  Weak Accept**

**Confidence:** 4

**Review:**

Paper Summary.
This works presents a method to map visual features from video streams into social tokens that can augment existing text tokens in LLMs to ground model predictions in social interactions and improve its performance in that setting. Particularly, their approach involves using a DINOv2 vision encoder to project the visual frames onto the LLMs embedding space creating both local and global social tokens to be combined with the input text token stream fed to the target LLM model (Gemma 2b in their experiments). Their finetuning methods focuses on the encoder part only (DINOv2) making it a lightweight method, as well as being scalable to other modalities (e.g., audio) as claimed by the authors. The experiments demonstrate improved perplexity results on social interaction datasets compared to when not having the social tokens.

Overall Assesment.
The method presented is lightweight, compatible with existing LLMs, and scalable to other modalities like audio. The paper has shown the method's usefulness for the scope of social interactions -- a specialized use-case to which the visual encoder was fine-tuned -- demonstrating significant improvements in perplexity for the target social datasets. However, it remains unclear whether/how injecting social tokens would impact the performance of the model on other tasks, which can be related to the Context Rot phenomena where performance can drop as more input tokens are injected to an LLM. Moreover, the work misses details on the training recipe and the encoder overhead in relation to the Gemma-2 base model. Still, I believe the approach possess merit as it addresses a relevant problem, and it aligns with the themes of the UniReps workshop. Hence, I'm leaning to accept.

Originality.
The method uses established building blocks (Gemma, DINOv2), though it is original in how it puts them together to mapvisual features as social tokens, taking frames associated with different nouns/verbs in text and projecting them as local/global social tokens to be combined with the input text

Significance.
Social grounding is an important problem for LLMs. Though the significance of the method would depend on how much the model retains its performance on broader class categories.

Quality.
The paper quality is good as an extended abstract - the authors state what needs to be done in future research to corroborate the claims

Strengths:
- The paper is easy to follow and clear in its objective and method
- Though it is just one form of evaluation and one model architecture, the perplexity results offer significant improvement higlighting the effectiveness of social tokens
- The method is simple, scalable and effective which is a positive for practical applicability

Weakness:
- My biggest concern is Context Rot - as you add tokens you risk losing performance in LLMs. I would've liked to see the impact of model performance on other settings not involving social interaction
- Need to see the experiments on other model architectures and sizes as the Gemma-2b represents one class of small models that may be particularly suboptimal for the target datasets
- An analysis of the overhead of the encoder, projection and processing with social tokens is missing (e.g., added inference latency).
- More training details for the encoder and projector in terms of GPU hours and configurations are needed to assess training overheads
- Qualitative examples from the datasets showcasing the performance of the model with and without the social tokens would help

**Score:**

3

**Topic Fit:**

2

---

### Official Review · Reviewer_f7a1 · 2025-09-12
**A Preliminary Look at Social Tokens in LLMs**

**Confidence:** 3

**Review:**

This paper introduces social tokens as a lightweight mechanism for grounding LLMs and VLMs in social interactions. Social tokens are derived from video frames using a fine-tuned vision encoder and projected into the LLM’s embedding space, where they are inserted alongside text tokens (locally after nouns/verbs and globally at the sequence start). This approach is simple yet effective: it reduces perplexity on held-out transcripts of social dialogues and captions, and attention analysis suggests that the tokens are actively leveraged by the model.

The method is clearly described, the motivation is strong, and the implications are compelling. Compared to heavier approaches like RLHF, this technique is lightweight and interpretable, making it a promising direction for socially aware multimodal models.

I have no major concerns given the scope of an extended abstract. While broader evaluation and ablations would be needed for a full paper, this is a good preliminary investigation into social tokens, showing they enhance prediction across entire transcripts. The work is clear, well-motivated, and raises interesting directions for socially grounding LLMs.

**Score:**

4

**Topic Fit:**

3

---

### Official Review · Reviewer_2CdY · 2025-09-16
**This paper introduces social tokens. These are lightweight embeddings derived from fine-tuned visual encoders.  They are projected into a frozen LLM to provide socially grounded visual context.**

**Confidence:** 1

**Review:**

The paper addresses the challenge of grounding LLMs in socially relevant multimodal cues, a space where current text-only and VLMs perform poorly. The authors propose social tokens, embeddings derived from visual encoders fine-tuned on videos of social interactions. These tokens are projected via an MLP into the LLM’s embedding space and inserted into the token stream. The LLM remains frozen, while the visual encoder and projector are optimized.

Strengths:
- Novelty: Social tokens offer a simple yet effective way to align visual social cues with frozen LLMs.
- Empirical results: Clear perplexity reduction on two socially relevant benchmarks.

Weaknesses:
- Broader validation would strengthen claims (more than two datasets).
- Only relying on POS-tag heuristics.
- Finetuning objectives can be described in bit more detail

**Score:**

3

**Topic Fit:**

3